# Fewer is More: A Deep Graph Metric Learning Perspective Using Fewer Proxies

**Yuehua Zhu**[1]    **Muli Yang**[1]    **Cheng Deng**[1]*    **Wei Liu**[2]

[1]School of Electronic Engineering, Xidian University, Xian, China
[2]Tencent AI Lab, Shenzhen, China

`{yuehuazhu, mlyang}@stu.xidian.edu.cn,`
`chdeng@mail.xidian.edu.cn, wl2223@columbia.edu`

## Abstract

Deep metric learning plays a key role in various machine learning tasks. Most of the previous works have been confined to sampling from a mini-batch, which cannot precisely characterize the global geometry of the embedding space. Although researchers have developed proxy- and classification-based methods to tackle the sampling issue, those methods inevitably incur a redundant computational cost. In this paper, we propose a novel Proxy-based deep Graph Metric Learning (ProxyGML) approach from the perspective of graph classification, which uses fewer proxies yet achieves better comprehensive performance. Specifically, multiple global proxies are leveraged to collectively approximate the original data points for each class. To efficiently capture local neighbor relationships, a small number of such proxies are adaptively selected to construct similarity subgraphs between these proxies and each data point. Further, we design a novel reverse label propagation algorithm, by which the neighbor relationships are adjusted according to ground-truth labels, so that a discriminative metric space can be learned during the process of subgraph classification. Extensive experiments carried out on widely-used CUB-200-2011, Cars196, and Stanford Online Products datasets demonstrate the superiority of the proposed ProxyGML over the state-of-the-art methods in terms of both effectiveness and efficiency. The source code is publicly available at `https://github.com/YuehuaZhu/ProxyGML`.

## 1   Introduction

Deep metric learning (DML) has been extensively studied in the past decade due to its broad applications, *e.g.*, zero-shot classification [37, 41, 36], image retrieval [35, 3], person re-identification [7, 48], and face recognition [38]. The core idea of DML is to learn an embedding space, where the embedded vectors of similar samples are close to each other while those of dissimilar ones are far apart from each other.

An embedding space with such a desired property is typically learned by metric losses, such as contrastive loss [19, 38] and triplet loss [8]. However, these losses rely on pairs or triplets constructed from samples in a mini-batch, empirically suffering from the *sampling issue* [7, 23] and leading to a polynomial growth with respect to the number of training examples. It thus turns out that the previous metric losses are highly redundant and less informative. In light of this, many efforts have been devoted to developing efficient hard/semi-hard negative sample mining strategies [7, 39] for handling the sampling issue. Essentially, these strategies still select hard samples from a subset (mini-batch) of the whole training data set, which fail to characterize the global geometry of the embedding space precisely.

Another type of methods circumvent such a sampling issue with a global consideration. For instance, ProxyNCA [23] assigns a trainable reference point to each class, namely *proxy*, and enforces each raw data point to be close to its relevant positive proxy and far away from the other negative proxies. During training, all the proxies are kept in the memory, therefore avoiding the sampling issue over different mini-batches. However, one proxy for each class is insufficient to represent complex intra-class variations (*e.g.*, poses and shapes of images). In view of this, MaPML [27] proposes to learn latent examples with different distortions to address various uncertainties in real world. On the other hand,

training with classification-based losses [21, 33, 34, 26] can also avoid the sampling issue by directly fitting the class distribution with fully-connected classification layers. However, the aforementioned methods *equally* treat each raw data point by calculating with either all reference points or class-specific parameters in classification layers, hence failing to capture the most discriminative relationships among raw data points. In addition, what follows is expensive computational consumption when many classes are involved [7].

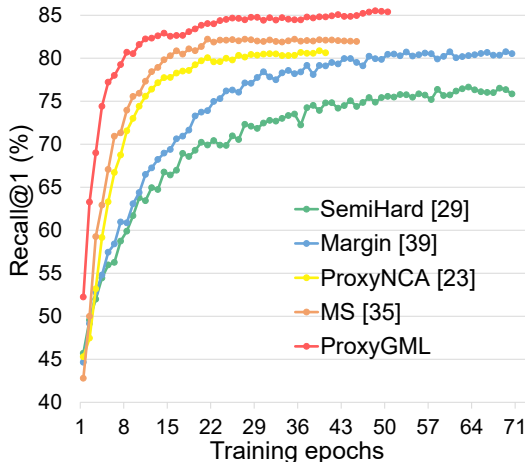

Figure 1: ProxyGML converges faster with higher Recall@1 values on the Cars196 test set (embedding dimension is 512 for all compared methods).

In this paper, we propose a novel Proxy-based deep Graph Metric Learning approach, dubbed ProxyGML, which uses fewer proxies to achieve better comprehensive performance (see Fig. 1) from a graph classification perspective. First, in contrast to ProxyNCA [23], we represent each class with multiple trainable proxies to better characterize the intra-class variations. Second, a directed similarity graph is constructed to model the global relationships between all the proxies and raw data samples in a mini-batch. Third, in order to capture informative fine-grained neighborhood structures for each raw data point, the directed similarity graph is decomposed into a series of $k$-nearest neighbor subgraphs by adaptively selecting a small number of informative proxies. Fourth, these subgraphs are classified according to their corresponding sample labels. In particular, inspired by the idea of label propagation (LP), we design a novel reverse LP algorithm to adjust the neighbor relationships in each subgraph with the help of known labels. Above all, the proxies and subgraphs collaborate to capture both global and local similarity relationships among raw data samples, so that a discriminative metric space can be learned in a both effective and efficient manner.

The major contributions of this paper are three-fold:

- We propose a novel reverse label propagation algorithm, offering a new insight into DML. To the best of our knowledge, this work firstly introduces graph classification into supervised DML.

- The proposed ProxyGML is an efficient drop-in replacement for existing DML losses, which can be readily applied to various tasks, such as image retrieval and clustering [9].

- Extensive experiments demonstrate the superiority of the proposed ProxyGML over the state-of-the-art methods in terms of both effectiveness and efficiency.

## 2    Related Work

**Distance-Based Deep Metric Learning.** Distance-based DML directly optimizes sample margins with conventional metric losses (*e.g.*, contrastive/triplet losses), suffering from the sampling issue [7, 23] and heavily requiring informative sample pairs for fast convergence. To seek informative pairs, Chopra *et al*. [2] introduced a contrastive loss which discards negative pairs whose similarities are smaller than a given threshold. Also, a hard sample mining strategy is proposed to find the most informative negative examples via an improved triplet loss [7]. N-Pair loss [30] and lifted structure loss [25] introduce new weighting schemes by designing a smooth weighting function to obtain more informative pairs. Alternatively, manifold proxy loss [1] is practically an extension of N-Pair

loss using proxies, and improves the performance by adopting a manifold-aware distance metric with heavy backbone ensembles. Besides, ProxyNCA [23] generates a set of proxies and optimizes the distances between each raw data sample and a full set of proxies, avoiding the sampling issue. Moreover, [35] introduces a multi-similarity (MS) loss with a general pair weighting strategy, which casts the sampling issue into a unified view of pair weighting by gradient analysis. Unlike the above methods, in this paper, we propose to leverage an easy-to-optimize graph-based classification loss to adjust the similarity relationships between each raw sample and fewer informative proxies.

**Classification-Based Deep Metric Learning.** Classification-based DML uses a classification layer to fit the distribution of each class [38]. To this end, plenty of classification-based methods (*e.g.*, center loss [38] and large-margin softmax loss [21]) have been developed to improve the feature discriminability. Specifically, center loss minimizes the distance between each raw data point and its class center, forming a class-dependent constraint. Large-margin softmax loss has been significantly improved by several recent types of losses such as additive margin softmax loss [33] and large-margin cosine loss [34]. Moreover, a method known as SoftTriple [26] utilizes multiple fully-connected layers that compute the similarities among features to classify each data sample, which can be viewed as an ensemble of multiple weak classifiers to improve the performance. In contrast, this work presents a novel graph-based reverse label propagation algorithm rather than classification layers to encode each sample's predictive output.

**Graph and Label Propagation.** A graph is basically composed of a set of nodes which are connected by edges. It is widely used to model pairwise relations between data objects or samples, which is good at capturing overall neighborhood structure [43] and possibly underlying manifold structure [20, 4]. Label propagation (LP) [47, 17, 18, 6, 15, 42, 16] is arguably the most popular algorithm for graph-based semi-supervised learning. LP is a simple yet effective tool, iteratively determining the unknown labels of samples according to appropriate graph structures [17, 22]. Inspired by the above methods, we design a variant of LP algorithm for DML to adjust the neighbor relations of graph nodes with the help of known labels.

## 3 Proposed Approach

This section describes the proposed ProxyGML approach. As shown in Fig. 2, ProxyGML contains three parts, *i.e.*, relation-guided graph construction, reverse label propagation, and classification-based optimization, which will be respectively elaborated below.

### 3.1 Formulation

Given a labeled training set with $C$ classes, our goal is to fine-tune a deep neural network towards yielding a more discriminative feature embedding. We adopt an episodic classification-based paradigm to achieve this goal, where $M$ samples in a mini-batch are randomly selected from the training data set in each episode. Denoting the embedding vector of the $i$-th data sample as $\mathbf{x}_i^s$ and its corresponding label as $y_i^s$, respectively, the embedding output of mini-batch samples extracted by a deep neural network can be defined as $\mathcal{S} = \{(\mathbf{x}_1^s, y_1^s), (\mathbf{x}_2^s, y_2^s), ..., (\mathbf{x}_M^s, y_M^s)\}$. Besides, we assign $N$ *trainable proxies* to each class, which can also be regarded as $N$ local cluster centers. The proxy set can be denoted by $\mathcal{P} = \{(\mathbf{x}_1^p, y_1^p), (\mathbf{x}_2^p, y_2^p), ..., (\mathbf{x}_{C \times N}^p, y_{C \times N}^p)\}$. We also denote the proxy labels in set $\mathcal{P}$ as a one-hot label matrix $\mathbf{Y}^p \in \{1, 0\}^{(C \times N) \times C}$ with $\mathbf{Y}_{ij}^p = 1$ if $y_i^p = j$ and $\mathbf{Y}_{ij}^p = 0$ otherwise. As shown in Fig. 2, during training, all the proxies in $\mathcal{P}$ and the mini-batch embedding $\mathcal{S}$ are fed into our proposed ProxyGML as input per iteration.

### 3.2 Relation-Guided Graph Construction

The fundamental philosophy behind DML is to ensure each data sample to be close to its relevant positive proxies and far away from its negative ones. Considering that graphs excel at modeling global data affinity, we propose to characterize overall neighbor relationships among samples with graphs. These graphs immediately serve for the proposed reverse LP, during which the label information of proxy nodes will be passed to sample nodes with certain probability scores according to adjacent similarities.

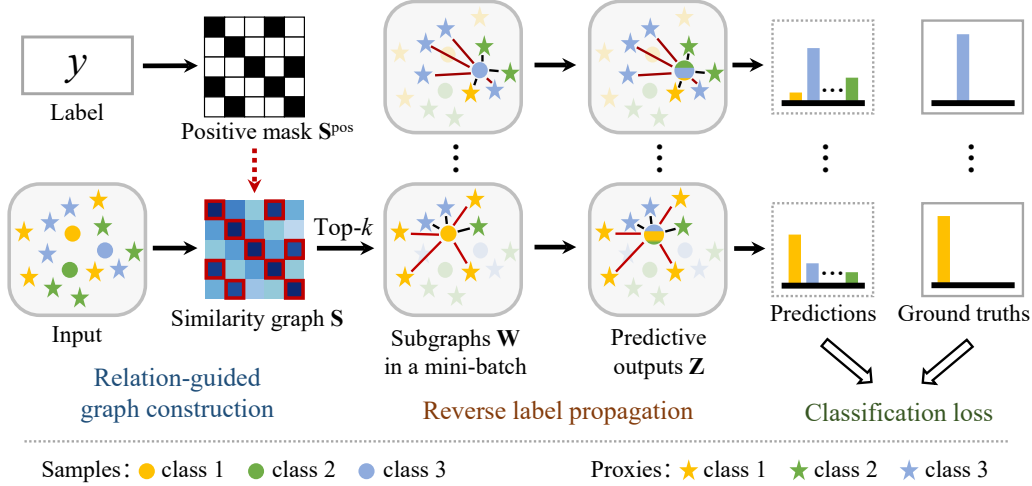

Figure 2: The pipeline of our proposed ProxyGML. The input contains samples in a mini-batch and all proxies, which serve as nodes in the directed graph. Different colors indicate different class labels. A multi-color sample indicates that its categorical information is affected by its neighboring proxies.

**Generating A Directed Similarity Graph.** Given embedding vectors $\{\mathbf{x}_i^s\}_{i=1}^{M}$ in a mini-batch and all proxies $\{\mathbf{x}_j^p\}_{j=1}^{C \times N}$, a directed graph is constructed, in which each node represents a sample or proxy, and each edge weight represents the similarity between two connected nodes. To measure the similarity relationships between samples and proxies, a common choice [44] in manifold learning and graph-based learning [20] is Gaussian similarity:

$$\mathbf{S}_{ij}^{\text{Gaussian}} = \exp\left(-\frac{\left(d(\mathbf{x}_i^s, \mathbf{x}_j^p)\right)^2}{2\sigma^2}\right), \tag{1}$$

where $d(\cdot, \cdot)$ denotes a distance measure (*e.g.*, Euclidean distance) and $\sigma$ is the length-scale hyperparameter. The neighborhood structure behaves differently with respect to various $\sigma$, making it nontrivial to select the optimal $\sigma$. To this end, we directly use cosine similarity to efficiently capture the relationship between sample $\mathbf{x}_i^s$ and proxy $\mathbf{x}_j^p$:

$$\mathbf{S}_{ij} = (\mathbf{x}_i^s)^\top \mathbf{x}_j^p, \tag{2}$$

where $\mathbf{S} \in \mathbb{R}^{M \times (C \times N)}$, both $\mathbf{x}_i^s$ and $\mathbf{x}_j^p$ are normalized to be unit length, and thus $\mathbf{S}_{ij} \in [-1, 1]$.

**Constructing $k$-NN Subgraphs.** With the generated similarity graph $\mathbf{S}$ over a mini-batch, the local relationships around each sample can be further constructed into a series of subgraphs, which help better capture the fine-grained neighborhood structures. A common way is keeping the $k$-max values in each row of $\mathbf{S}$ to construct $k$-nearest neighbor ($k$-NN) subgraphs. Particularly, in our setting, the proxies serve as 1) positive reference points for each class-corresponding sample and 2) cluster centers in each class. Since all proxies are randomly initialized, directly selecting $k$-nearest proxies for each sample will be likely to miss plenty of positive proxies, so that these proxies of the same class cannot be simultaneously updated per iteration. While as randomly initialized cluster centers, proxies of the same class should be close to each other, *i.e.*, all of them should be included for optimization to guarantee that. In light of this, we introduce a positive mask $\mathbf{S}^{\text{pos}}$ to ensure that all positive proxies for each sample are selected, and its validity will be proven in Sec. 4.2. $\mathbf{S}^{\text{pos}}$ can also be regarded as a "soft" constraint on proxies, which makes similar proxies mutually close by encouraging proxies to be close to their relevant samples.

The positive mask $\mathbf{S}^{\text{pos}}$ is derived from the labels of samples and proxies, which inherently reflects the authentic similarity relationships between them:

$$\mathbf{S}_{ij}^{\text{pos}} = \begin{cases} 1, & \text{if } y_i^s = y_j^p, \\ 0, & \text{else.} \end{cases} \tag{3}$$

Under the guidance of the positive mask, we calculate and store the indexes of $k$-max values in each row of $(\mathbf{S} + \mathbf{S}^{\text{pos}})$ into a $k$-element set $\mathcal{I} = \{(i, j), \cdots\}$. Then the subgraphs are constructed and

represented by a sparse neighbor matrix $\mathbf{W}$:

$$\mathbf{W}_{ij} = \begin{cases} \mathbf{S}_{ij}, & \text{if } (i,j) \in \mathcal{I}, \\ 0, & \text{else,} \end{cases} \tag{4}$$

where $\mathbf{W} \in \mathbb{R}^{M \times (C \times N)}$. As shown in Fig. 2, with the aid of the positive mask $\mathbf{S}^{\text{pos}}$, all the positive proxies of each sample will be involved in each constructed subgraph even with a relatively small $k$.

Specifically, $k$ is given by $k = \lceil r \times C \times N \rceil$, where we introduce $r \in (0,1]$ to expediently obtain subgraphs at different scales, and $\lceil \cdot \rceil$ is a ceiling function to ensure that $k$ is an integer.

## 3.3 Reverse Label Propagation

Each constructed subgraph actually reflects a manifold structure, where data points from the same category should be close to each other [11]. Recall that the idea behind traditional LP in semi-supervised learning is to infer unknown labels by virtue of manifold structure [20]. On the contrary, we seek to leverage known labels to adjust the manifold structure using the proposed *reverse label propagation (RLP)* algorithm. Concretely, we first encode all subgraphs $\mathbf{W}$ into predictive outputs $\mathbf{Z}$ following the idea of original LP:

$$\mathbf{Z} = \mathbf{W}\mathbf{Y}^p, \tag{5}$$

where $\mathbf{Z} \in \mathbb{R}^{M \times C}$ actually reflects that how categorical information of a sample is influenced by its neighboring proxies.

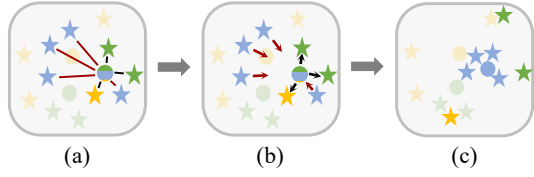

(a)        (b)        (c)

Figure 3: Subgraph evolution during the loss back-propagation process. Meanings of colors and shapes are the same as Fig. 2.

Fig. 3 illustrates how manifold structure evolves when $\mathbf{Z}$ is optimized with the classification loss. The manifold structure in Fig. 3(a) contains a sample and its corresponding adaptively selected seven proxies, *i.e.*, four positive proxies (guided by $\mathbf{S}^{\text{pos}}$) and three negative proxies. The categorical information of this sample is thus affected by the seven proxies. After back-propagating with the classification loss (*cf*. Sec 3.4 for a detailed discussion), the positive proxies will be pulled closer to this sample, while the negative ones will be pushed farther away, as shown in Fig. 3(b). As a result, we can expect a favorable manifold structure as shown in Fig. 3(c), such that all samples are eventually surrounded by their corresponding positive proxies, *i.e.*, local cluster centers. This result is in accordance with the goal of DML.

## 3.4 Classification-Based Optimization

Now we consider what happens during the classification learning process. As shown in Fig. 2, the predictive outputs $\mathbf{Z}$ are first converted to prediction scores $\mathbf{P}$ by softmax operation, and are then optimized to fit the one-hot ground-truth label distribution. Consequently, each element of $\mathbf{P}$, which reflects the cumulative similarity between a sample and positive or negative proxies, will be either amplified or suppressed, corresponding to the pull or push operation in Fig. 3(b).

**Classification Loss on Raw Samples.** Practically, $\mathbf{Z}$ is highly sparse due to small $k$, and many zero entries in $\mathbf{Z}$ will result in an inflated denominator in a traditional softmax function, which cannot correctly encode the subgraph predictions. Therefore, we propose a novel mask softmax function to prevent zero values from contributing to the prediction scores:

$$P(\tilde{y}_i^s = j | \mathbf{x}_i^s) = \frac{\mathbf{M}_{ij}\exp(\mathbf{Z}_{ij})}{\sum\limits_{j'=1}^{C} \mathbf{M}_{ij'}\exp(\mathbf{Z}_{ij'})}, \tag{6}$$

where $\tilde{y}_i^s$ denotes the predicted label for the $i$-th sample $\mathbf{x}_i^s$ in $\mathcal{S}$, $\mathbf{Z}_{ij}$ denotes the $j$-th predictive element for the $i$-th sample, and mask $\mathbf{M} \in \{1,0\}^{M \times C}$ with $\mathbf{M}_{ij}=0$ if $\mathbf{Z}_{ij}=0$ and $\mathbf{M}_{ij}=1$ otherwise.

The cross-entropy loss between prediction scores and ground-truth labels over each sample is calculated in an end-to-end fashion:

$$\mathcal{L}^s = -\frac{1}{M} \sum_{i=1}^{M} \sum_{j=1}^{C} \mathbb{I}(y_i^s = j) \log\left(P(\tilde{y}_i^s = j | \mathbf{x}_i^s)\right), \tag{7}$$

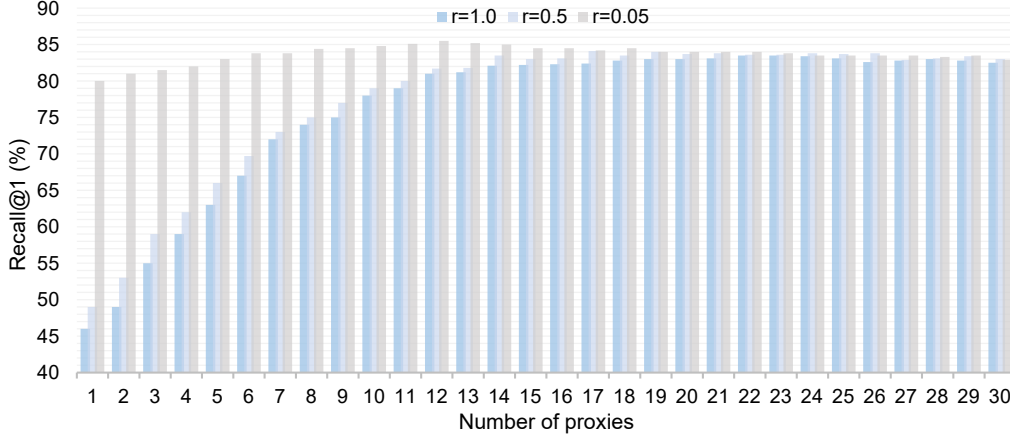

Figure 4: Recall@1 values on Cars196 with different numbers of proxies $N$ under three different $r$.

where $y_i^s$ denotes the ground-truth label of $\mathbf{x}_i^s$, and $\mathbb{I}(b)$ is an indicator function with $\mathbb{I}(b) = 1$ if $b$ is true and $\mathbb{I}(b) = 0$ otherwise.

**Regularization on Proxies.** Since positive proxies serve as local cluster centers in each class, we therefore impose a "hard" constraint on the proxies to ensure that similar proxies are close to each other while dissimilar ones are far apart from each other. To be specific, we regard each proxy as a "sample" and the other similar/dissimilar proxies as "positive/negative proxies" with regard to this "sample". Following Section 3.2, we can construct a similarity graph between the proxies:

$$\mathbf{S}_{ij}^p = (\mathbf{x}_i^p)^\top \mathbf{x}_j^p, \tag{8}$$

where $\mathbf{S}^p \in \mathbb{R}^{(C \times N) \times (C \times N)}$, and both $\mathbf{x}_i^p$ and $\mathbf{x}_j^p$ are normalized to be unit length. Because the global geometry of randomly initialized proxies is computationally expensive to preserve, we do not further construct $k$-NN subgraphs for $\mathbf{S}^p$. Therefore, according to reverse LP, the predictive outputs of those "samples" are derived as follows,

$$\mathbf{Z}^p = \mathbf{S}^p \mathbf{Y}^p. \tag{9}$$

The outputs $\mathbf{Z}^p$ can also be converted to the prediction scores:

$$P(\tilde{y}_i^p = j | \mathbf{x}_i^p) = \frac{\exp(\mathbf{Z}_{ij}^p)}{\sum\limits_{j'=1}^{C} \exp(\mathbf{Z}_{ij'}^p)}, \tag{10}$$

where $\tilde{y}_i^p$ denotes the predicted label for the $i$-th proxy $\mathbf{x}_i^p$ in $\mathcal{P}$. Then the cross-entropy loss over each proxy is computed as:

$$\mathcal{L}^p = -\frac{1}{C \times N} \sum_{i=1}^{C \times N} \sum_{j=1}^{C} \mathbb{I}(y_i^p = j) \log\big(P(\tilde{y}_i^p = j | \mathbf{x}_i^p)\big). \tag{11}$$

With this regularization on proxies, our ultimate objective becomes

$$\mathcal{L}(\Theta, \mathcal{P}) := \mathcal{L}^s + \lambda \mathcal{L}^p, \tag{12}$$

where $\Theta$ denotes the parameters of a backbone network responsible for feature embedding, $\mathcal{P}$ is the desired proxy set, and $\lambda > 0$ is the trade-off hyper-parameter. An end-to-end training by minimizing our objective $\mathcal{L}$ yields a discriminative metric space and the most informative proxies. It is noted that $\lambda$ will be shown insensitive to the final performance in our experiments. A detailed sensitivity test for $\lambda$ is given in the supplementary material.

## 4 Experiments

In this section, we evaluate our proposed ProxyGML on three widely-used benchmarks for both image clustering and image retrieval tasks.

### 4.1 Experimental Setup

**Datasets.** Experiments are conducted on CUB-200-2011 [32], Cars196 [14], and Stanford Online Products [25] datasets. We follow the conventional protocol [25, 40, 26] to split them into training and test parts.

*CUB-200-2011* [32] covers 200 species of birds with 11,788 instances, where the first 100 species (5,864 images) are used for training and the rest 100 species (5,924 images) for testing.

*Cars196* [14] is composed of 16,185 car images of 196 classes. We use the first 98 classes (8,054 images) for training and the other 98 classes (8,131 images) for testing.

*Stanford Online Products* [25] contains 22,634 classes with 120,053 product images in total, where the first 11,318 classes (59,551 images) are used for training and the remaining 11,316 classes (60,502 images) are used for testing.

**Evaluation Metrics.** Following the standard protocol [30, 25], we calculate Recall@$n$ on the image retrieval task. Specifically, for each query image, top-$n$ nearest images are returned based on Euclidean distance, and then the recall score will be calculated by treating the images sharing the same class label as the query positive (*i.e.*, relevant) and the others negative (*i.e.*, irrelevant). For clustering evaluation, we adopt the $K$-means clustering algorithm to cluster instances and the clustering quality is reported in Normalized Mutual Information (NMI). Both Recall@$n$ and NMI are measured on the test set of any dataset for all experiments.

**Implementation Details.** Our method is implemented in PyTorch with an NVIDIA TITAN XP GPU of 12GB memory. All input images are resized to $224 \times 224$. For data augmentation, we perform standard random cropping and horizontal mirroring for training instances, while a single center cropping for testing instances. Following SoftTriple [26] and MS [35], we employ Inception [10] pre-trained on the ImageNet [28] dataset as our backbone feature embedding network with the embedding dimension as $512$. Similar to ProxyNCA [23], we only use a small mini-batch size $M$ of 32 images. The model is optimized by Adam [13] within 50 epochs. The initial learning rates for the backbone and ProxyGML (trainable proxies) are respectively set to $1e-4$ and $3e-2$, decreasing by $0.1$ every 20 epochs. The number of proxies $N$ and the ratio $r$ for determining $k$ are set to 12 and $0.05$, respectively, unless expressly stated. The regularizer weight $\lambda$ is not sensitive and we empirically set it to $0.3$. Note that each class of the Stanford Online Products dataset merely has 5 images in average, so we set $N = 1$ for determining $k$ on this dataset without the regularizer, and the initial learning rate for trainable proxies is increased from $3e-2$ to $3e-1$. In particular, ProxyGML can be regarded as a DML loss and all proposed modules with proxies will be totally removed during the testing phase.

### 4.2 Parameter Analysis and Ablation Study

To evaluate the efficacy of our proposed ProxyGML, we investigate the impact of the number of selected proxies $k$ (which is actually controlled by $N$ and $r$), the effectiveness of the positive mask in Eq. (3), the mask softmax in Eq. (6), and the regularizer in Eq. (11).

**Impact of $N$.** As discussed in Sec. 3.2, $k$ determines the size of $k$-NN subgraphs. While the upper bound of $k$ is $C \times N$, we expect to select a relatively small $k$ in consideration of both effectiveness and efficiency. Thus, $r$ is introduced to directly select $k$ at different scales. Empirically, we experiment on three representative scales, *i.e.*, $r = 0.05$, $0.5$, and $1$, to explore the impact of $N$, as shown in Fig. 4. In general, over all three different $r$, the best performance is achieved when $N = 12$, which confirms that the learned feature embedding can better capture intra-class variations with a proper number of local cluster centers. When $N$ further increases, the performance degrades due to overfitting when the proxies are over-parameterized. Notably, it can be seen in Fig. 4 that a smaller size of nearest neighbor graph ($r = 0.05$) can provide more stable and better performance, which will be further discussed below. Also, an exploration of broader combinations of $r$ and $N$ is presented in the supplementary material.

**Impact of $r$.** We study the effect of $r$ with $N$ fixed to 12. As shown in Fig. 5, the best performance is achieved when $r = 0.05$, which suggests that the proposed ProxyGML can select a small number of representative proxies for each sample to construct discriminative subgraphs. When $r < 0.05$, fewer proxies (especially the negative ones, with the effect of the positive mask) are involved in the subgraphs, causing the inferior performance. On the other hand, when $r > 0.05$, the subgraphs

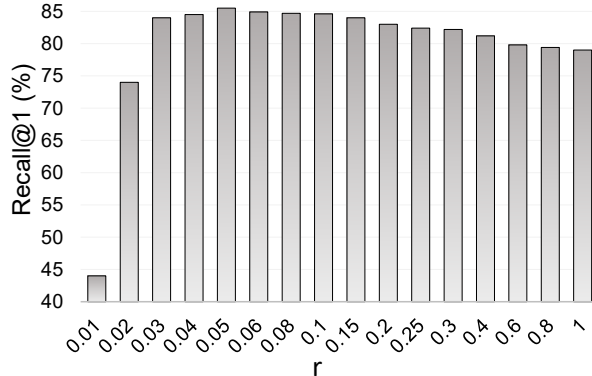

Figure 5: Recall@1 values of ProxyGML on the Cars196 dataset with different sizes of neighbor graphs (controlled by $r$) when $N = 12$.

contain redundant negative proxies, also leading to the unfavorable performance. Although the total number of proxies in our method is up to $C \times N$, only a few proxies are adaptively selected for different samples, which not only reduces the computational consumption but also captures more informative neighborhood structures to boost the training quality.

**Impact of $\mathbf{S}^{\text{pos}}$, $\mathbf{M}$ and $\mathcal{L}^p$.** We ablate our proposed ProxyGML to evaluate the effectiveness of the positive mask $\mathbf{S}^{\text{pos}}$ in Eq. (3), the mask softmax function (indicated by $\mathbf{M}$) in Eq. (6), and the regularizer $\mathcal{L}^p$ on proxies in Eq. (11). As shown in Table 1, the ablation is eight-fold: #1 is the base loss, *i.e.*, $\mathcal{L}^s$ with original softmax, without any of the three proposed modules; in #2–4, each of the modules is added into the base loss; in #5–7, two of the three modules are introduced into the base loss; #8 is the full loss.

As demonstrated in Table 1, each proposed module contributes to the overall performance of our ProxyGML. Specifically, the positive mask $\mathbf{S}^{\text{pos}}$ helps construct effective subgraphs and encourages to learn better cluster centers, which significantly improves the overall performance. The mask softmax function $\mathbf{M}$ outperforms the traditional softmax by preventing zero values from contributing to the prediction scores, and hence improves the classification optimization, which mainly benefits the image retrieval task. Additionally, the regularizer $\mathcal{L}^p$ on proxies further improves the performance by regularizing the global geometry among proxies, thereby enabling to learn more discriminative proxies. To summarize, full ProxyGML produces the best result, which validates the effectiveness of all the proposed modules.

Table 1: The ablation study for three different modules on Cars196.

| # | $\mathbf{S}^{\text{pos}}$ | $\mathbf{M}$ | $\mathcal{L}^p$ | NMI | R@1 |
|---|---|---|---|---|---|
| 1 | × | × | × | 52.1 | 47.3 |
| 2 | ✓ | × | × | 69.6 | 83.3 |
| 3 | × | ✓ | × | 54.9 | 66.1 |
| 4 | × | × | ✓ | 67.1 | 81.7 |
| 5 | × | ✓ | ✓ | 68.8 | 82.6 |
| 6 | ✓ | × | ✓ | 71.6 | 84.5 |
| 7 | ✓ | ✓ | × | 70.7 | 84.0 |
| 8 | ✓ | ✓ | ✓ | 72.4 | 85.5 |

### 4.3 Comparison with State-of-the-Arts

We compare ProxyGML against three types of methods including:

1) *Sampling-based methods*, *i.e.*, SemiHard [29], LiftedStruct [25], HDC [45], and HTL [5];
2) *Clustering-based methods*, *i.e.*, Clustering [24] and ProxyNCA [23];
3) *Other recent methods*, *i.e.*, DAMLRRM [40], HDML [46], MS [35], and SoftTriple [26].

Table 2 reports the clustering and retrieval results of ProxyGML and those of all above competitors on CUB-200-2011, Cars196, and Stanford Online Products, respectively. For fair comparison, we report the performance of ProxyGML with varying embedding dimension in $\{64, 384, 512\}$. As exhibited in Table 2, ProxyGML generally outperforms the state-of-the-art methods on the three benchmark datasets. Notably, ProxyGML does not consistently outperform the most competitive baselines on Stanford Online Products under all metrics. The main reason is that this dataset contains a huge number of classes ($11,318$ classes) with a low intra-class variance, *i.e.*, each class contains 5 images in average, which goes against the advantage of multiple local cluster centers. However, ProxyGML achieves comparable results with a much less computational cost. Specifically, MS [35]

Table 2: Comparison with the state-of-the-art methods. The performances of clustering and retrieval are respectively measured by NMI (%) and Recall@$n$ (%). Superscript denotes embedding dimension. "–" means that the result is not available from the original paper. Backbone networks are denoted by abbreviations: BN—Inception with batch normalization [10], G—GoogleNet [31].

| Method | | CUB-200-2011 | | | | Cars196 | | | | Stanford Online Products | | | |
|---|---|---|---|---|---|---|---|---|---|---|---|---|---|
| | | NMI | R@1 | R@2 | R@4 | NMI | R@1 | R@2 | R@4 | NMI | R@1 | R@10 | R@100 |
| SemiHard[29] | BN | 55.4 | 42.6 | 55.0 | 66.4 | 53.4 | 51.5 | 63.8 | 73.5 | 89.5 | 66.7 | 82.4 | 91.9 |
| Clustering[24] | BN | 59.2 | 48.2 | 61.4 | 71.8 | 59.0 | 58.1 | 70.6 | 80.3 | 89.5 | 67.0 | 83.7 | 93.2 |
| LiftedStruct[25] | G | 56.6 | 43.6 | 56.6 | 68.6 | 56.9 | 53.0 | 65.7 | 76.0 | 88.7 | 62.5 | 80.8 | 91.9 |
| ProxyNCA[23] | BN | 59.5 | 49.2 | 61.9 | 67.9 | 64.9 | 73.2 | 82.4 | 86.4 | 90.6 | 73.7 | – | – |
| HDC[45] | G | – | 53.6 | 65.7 | 77.0 | – | 73.7 | 83.2 | 89.5 | – | 69.5 | 84.4 | 92.8 |
| HTL[5] | BN | – | 57.1 | 68.8 | 78.7 | – | 81.4 | 88.0 | 92.7 | – | 74.8 | 88.3 | 94.8 |
| DAMLRRM[40] | G | 61.7 | 55.1 | 66.5 | 76.8 | 64.2 | 73.5 | 82.6 | 89.1 | 88.2 | 69.7 | 85.2 | 93.2 |
| HDML[46] | G | 62.6 | 53.7 | 65.7 | 76.7 | 69.7 | 79.1 | 87.1 | 92.1 | 89.3 | 68.7 | 83.2 | 92.4 |
| SoftTriple[26] | BN | 69.3 | 65.4 | 76.4 | 84.5 | 70.1 | 84.5 | 90.7 | 94.5 | **92.0** | 78.3 | 90.3 | 95.9 |
| MS[35] | BN | – | 65.7 | 77.0 | 86.3 | – | 84.1 | 90.4 | 94.0 | – | 78.2 | 90.5 | 96.0 |
| ProxyGML[64] | BN | 65.1 | 59.4 | 70.1 | 80.4 | 67.9 | 78.9 | 87.5 | 91.9 | 89.8 | 76.2 | 89.4 | 95.4 |
| ProxyGML[384] | BN | 68.4 | 65.2 | 76.4 | 84.3 | 70.9 | 84.5 | 90.4 | 94.5 | 90.1 | 77.9 | 90.0 | 96.0 |
| ProxyGML[512] | BN | **69.8** | **66.6** | **77.6** | **86.4** | **72.4** | **85.5** | **91.8** | **95.3** | 90.2 | 78.0 | **90.6** | **96.2** |

adopts a very large batch size 1000 (see its appendix) to achieve its best performance, which is difficult for us to reproduce even using four GPUs, each with 12 GB memory. Proxy-Anchor [12] also requires a large batch size 180 and compares each sample with all $11,318$ proxies; SoftTriple [26] employs two parallel FC layers to classify $11,318$ classes. In contrast, our method only needs to calculate and update the gradients of $k = \lceil 0.05 \times 11318 \times 1 \rceil$ proxies for each sample during back-propagation. And we use a small batch size 32 for *inheriting the advantage* of original ProxyNCA. More comparisons (concerning time, memory consumption, and newly proposed Proxy-Anchor [12]) are also provided in the supplementary material. Overall, our experiments demonstrate the superiority of the proposed ProxyGML in terms of both effectiveness and efficiency.

## 5   Conclusions

In this paper, we proposed a novel Proxy-based deep Graph Metric Learning (ProxyGML) approach from the perspective of graph classification, which offers a new insight into deep metric learning. The core idea behind ProxyGML is "fewer proxies yield more efficacy". By adaptively selecting the most informative proxies for different samples, ProxyGML is able to efficiently capture both global and local similarity relationships among the raw samples. Besides, the proposed reverse label propagation algorithm goes beyond the setting of semi-supervised learning. It allows us to adjust the neighbor relationships with the help of ground-truth labels, so that a discriminative metric space can be learned flexibly. The experimental results on CUB-200-2011, Cars196, and Stanford Online Products benchmarks demonstrate the superiority of ProxyGML over the state-of-the-arts.

**Acknowledgment**

Our work was supported in part by the National Natural Science Foundation of China under Grant 62071361 and the National Key R&D Program of China under Grant 2017YFE0104100.

**Broader Impact**

*a) Who may benefit from this research?* In this paper we proposed a new pipeline for deep metric learning. Like many other relevant studies in this area, our work aims at establishing similarity or dissimilarity relationships among data inputs. Our work can be applied to many practical scenarios, such as big data analysis, face/object recognition, person re-identification, voice verification, *etc*. Corporations or other non-profit organizations/persons with such purposes may benefit from our work. *b) Who may be put at disadvantage from this research?* Since our work can be used in social media

companies or any other occasions where user data can be accessed, people who are worried about their privacy being analyzed or targeted may be put at disadvantage. *c) What are the consequences of failure of the system?* Before formal deployment, the DML model should be properly trained and tested with available data samples, *i.e.*, the risk should be controllable. If any failure happens, the most immediate consequence can be recognition/analysis errors for the systems in which our proposed model is leveraged, which may further result in unnecessary economic costs or losses of other resources. *d) Whether the task/method leverages biases in the data?* Our work is posed with a general purpose of learning a more discriminative feature embedding without specific requirements on training data. Thus, our work does not leverage biases in the data, but rather, may possess the ability to suppress/capture such biases (if any) using our adaptive proxy strategy.

## Footnotes

*The corresponding author.

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
