[Supplementary Material]

# Supplementary Material for "Fewer is More: A Deep Graph Metric Learning Perspective Using Fewer Proxies"

**Yuehua Zhu**[1]    **Muli Yang**[1]    **Cheng Deng**[1]    **Wei Liu**[2]

[1]School of Electronic Engineering, Xidian University, Xian, China
[2]Tencent AI Lab, Shenzhen, China
{yuehuazhu, mlyang}@stu.xidian.edu.cn,
chdeng@mail.xidian.edu.cn, wl2223@columbia.edu

## 1   Comparison of Training Time and Memory Consumption

We report in Table 1 the comparison of training time and GPU memory consumption between our proposed ProxyGML and two types of state-of-the-art methods, *i.e.*, sampling-based Semi-Hard [6], Margin [10], HDML [11], MS [9], and proxy-based ProxyNCA [3]. Inception [7] pretrained on the ImageNet [5] dataset is employed as the backbone feature embedding network (embedding dimension is 512) for all the compared methods. The other parameters (*e.g.*, batch size) follow the default settings of these methods. All experiments are implemented with an NVIDIA TITAN XP GPU of 12GB memory.

Technically, ProxyGML adaptively selects a few proxies for each sample to construct informative $k$-NN subgraphs, which can be viewed as a novel sampling strategy in the proxy level. In contrast to other sampling-based methods, according to Table 1, ProxyGML iterates and converges faster with a less memory requirement. The main reason is two-fold: 1) ProxyGML selects the proxies using a simple ranking algorithm instead of the cumbersome sampling strategies in the sample level, and most of calculations in ProxyGML are also simple matrix/vector multiplications; 2) since proxies can collectively approximate the global geometry of raw data samples, a large batch size is unnecessary for ProxyGML; so ProxyGML converges fast even with a small batch size.

Specifically, compared against ProxyNCA [3], ProxyGML introduces a proxy sampling phase, which increases the iteration time; the extra uncertainties also increase the convergence time. In ProxyGML, multiple trainable proxies are assigned to each class, which also increases the memory consumption. We believe that the additional training time and memory requirement are worthy given the brought great gain in accuracy (*cf.* Table 2 in the original paper).

In conclusion, ProxyGML is more efficient than the aforementioned sampling-based methods, and we argue that sampling in the proxy level should be more promising than sampling in the sample level.

Table 1: Comparison of iteration time (training time per iteration), convergence time (training time till convergence), and maximum GPU memory consumption on the Cars196 dataset.

| Time/Memory | Semi-Hard [6] | Margin [10] | ProxyNCA [3] | HDML [11] | MS [9] | ProxyGML |
|---|---|---|---|---|---|---|
| Iteration time | 0.48 s | 0.56 s | 0.17 s | 1.1 s | 0.75 s | 0.23 s |
| Convergence time | 1.01 h | 1.12 h | 0.51 h | 2.25 h | 0.88 h | 0.81 h |
| Max GPU memory | 4.90 GB | 4.90 GB | 1.54 GB | 8.96 GB | 3.52 GB | 2.18 GB |

The codes are downloaded from

1) `https://github.com/Confusezius/Deep-Metric-Learning-Baselines` (Semi-Hard [6], Margin [10], and ProxyNCA [3]);

Figure 1: Recall@1 values of ProxyGML on Cars196 with different combinations of $N$ and $r$.

The datasets are available at

1) `https://ai.stanford.edu/~jkrause/cars/car_dataset.html` (Cars196 [2]);
2) `http://www.vision.caltech.edu/visipedia/CUB-200-2011.html` (CUB-200-2011 [8]);
3) `https://cvgl.stanford.edu/projects/lifted_struct/` (Stanford Online Products [4]).

## 2 Comparison with Proxy-Anchor [1]

We also compare our proposed ProxyGML against newly proposed Proxy-Anchor [1] using its official code, and the Recall@1 results are listed in Table 2. In particular, we have found that Proxy-Anchor relies on a large batch size, and is implemented with three additional engineering skills, *i.e.*, 1) a combination of an average- and a max- pooling layers following the Inception backbone, 2) a warm-up strategy for stabilizing proxy learning, and 3) an AdamW optimizer instead of original Adam. For fair comparison, we evaluate Proxy-Anchor under our setting — with batch size 32 and the three engineering skills removed; it is also evaluated with the three skills enabled (indicated by "*"),

Table 2: Comparison with Proxy-Anchor on the Cars196 dataset. The performance of image retrieval is measured by Recall@n (%).

| Method | CUB | Cars196 | SOP |
|---|---|---|---|
| ProxyGML$_{32}$ | **66.6** | **85.5** | **78.0** |
| Proxy-Anchor$_{32}$ | 35.8 | 20.3 | 41.4 |
| Proxy-Anchor$^*_{32}$ | 65.4 | 83.1 | 75.7 |
| Proxy-Anchor$_{180}$ | 66.1 | 84.2 | 54.5 |
| Proxy-Anchor$^*_{30}$ | 65.9 | 84.6 | 76.0 |

and with its optimal batch size 180. Since time does not allow any further tuning for Proxy-Anchor, we report here the result with batch size 30 (also the skills are used) provided in its paper for reference. Please note that this is only a preliminary experiment. Still, we can infer from the table the advantage of our ProxyGML over Proxy-Anchor. We will further conduct more experiments of Proxy-Anchor with a careful tuning and ProxyGML with large batch size and the three skills added, which will be available at `https://github.com/YuehuaZhu/ProxyGML`.

## 3   Sensitivity Test for $\lambda$

As shown in Fig. 2, the tradeoff hyper-parameter $\lambda$ imposed on the regularizer $\mathcal{L}^p$ on proxies is insensitive. Particularly, as demonstrated in Table 1 in the original paper, both the positive mask $\mathbf{S}^{\mathrm{pos}}$ and regularizer $\mathcal{L}^p$ are conducive to learning better proxies, *i.e.*, better local cluster centers in each class. Therefore, the presence or absence of the regularizer $\mathcal{L}^p$ will not greatly affect the overall performance when the positive mask $\mathbf{S}^{\mathrm{pos}}$ exists, so the tradeoff hyper-parameter $\lambda$ on $\mathcal{L}^p$ is insensitive.

Figure 2: Recall@1 values of ProxyGML on the Cars196 dataset with different $\lambda$.

## 4   Impact of Broader Combinations of $N$ and $r$

We show in Fig. 1 the impact of representative combinations of different $N$ and $r$ on Cars196 whose number of classes $C$ is 98. Specifically, for the $i$-th sample $(\mathbf{x}_i^s, y_i^s)$ in a mini-batch, its 98-dimensional prediction score vector can be derived from the $i$-th row of $\mathbf{Z}$ (Eq. (5) in the original paper) through a softmax operation. In fact, the value of $\mathbf{Z}_{ij}$ reflects the *cumulative similarity* between the sample $\mathbf{x}_i^s$ and the $j$-th class proxies, *i.e.*, $N$ positive proxies and $(\lceil r \times 98 \times N \rceil - N)$ negative proxies (*cf.* Sec. 3.2 in the original paper).

Now we consider two special cases. When $r = 0.01$, no negative proxies will be selected. In this case, negative elements in the prediction scores will all be zeros (*cf.* Fig. 2 in the original paper), *i.e.*, the prediction score corresponding to class $y_i^s$ will be equal to 1, such that the cross-entropy loss will be zero and the trainable parameters will not be updated at all, causing poor performance shown in Fig. 1. When $r = 1$ and $N = 1$, only 1 positive proxy will be selected while the number of negative ones is 97. After softmax operation, the prediction score corresponding to class $y_i^s$ will be restricted to far smaller than 1, making it hard to be optimized to fit the one-hot ground-truth label distribution and also leading to poor performance. Overall, we can observe the optimal performance when $r = 0.05$ and $N = 12$.