[Reviews · NeurIPS 2020]

Review 1

Summary and Contributions: Post Rebuttal: The author addressed my main concerns. So I would like to upgrade my score. ------------------ The author introduced Graph classification into DML. The experiments show improvement in several benchmark tasks.

Strengths: The results show improvement over recent baseline models. The experiment including the ablation study is sound on several datasets.

Weaknesses: The use of ground truth labels to compute S^{POS} will lead to an unbalance between training and testing since when testing, there should not be any ground truth labels. Even though the author claimed that they introduced graph classification into DML, the actual use is unclear. The current version seems only to use k-top neighbors, which will be a trivial graph. There is no message passing between nodes. The difference between the learnable proxies and some trivial methods like cluster is unclear.

Correctness: The claim of S^{POS} is unclear when testing.

Clarity: Yes

Relation to Prior Work: Yes

Reproducibility: Yes

Additional Feedback: 1. The biggest concern I have is the use of S^{POS}. 2. Another one is that the use of a graph is unnecessary. There seems to be no need to build the graph, other than the top-k neighbors. 3. For the proxy, they are initialed randomly. So different proxies may have the same representative. Is there any method to hard constraint this problem, or piratically this will not be an issue? A similar problem will happen when directly use k-NN for machine learning.


Review 2

Summary and Contributions: This paper introduces a new deep metric learning method, namely ProxyGML, which integrates graph classification into the embedding space learning. ProxyGML globally represents each class in the data using several proxies, and selects a few most informative ones to construct a series of sub-graphs that encodes the local similarity relations between proxies and samples. By classifying these sub-graphs with the ground-truth labels, a favorable embedding space can be learned through the proposed reverse label propagation. The proposed method is evaluated on image retrieval and clustering tasks on CUB-200-2011, Cars196, and Stanford Online Products datasets. Ablation studies and comparison with the state-of-the-arts have demonstrated the effectiveness of the proposed method.

Strengths: - Deep metric learning with graph classification is an interesting perspective, in which the proposed "reverse label propagation" algorithm is able to adaptively adjust the embedding manifold through a classification loss. I'm positive to see such novel idea in this area, which I assume is also of interest to the NeurIPS community. - The paper is well structured, and all the contributions are motivated well and clearly discussed; also all the information required to reproduce the results seems to be in place. - Experiments are comprehensive. The proposed method is compared against a series of state-of-the-arts, and evaluated in terms of time and memory costing. Also the parameters are well discussed through ablation study, which is convincing.

Weaknesses: - There are two important parameters in the proposed method, i.e., N, r. While the presented ablation studies are only conducted on Cars196 dataset, does it follow the same pattern on the other two datasets as on Cars196? Also, how can we choose an optimal combination of these parameters for a new dataset without exhaustive grid search? Is there any empirical experience? The authors are encouraged to give some more analysis into this. - The proposed method performs less favorable on Stanford Online Products datasets than on the other two involved datasets. The authors argued in lines 297--299 that low intra-class variance goes against the advantage of the proposed method. I understand that there is no free lunch, but is there any supplemental way to fix this or is the proposed method not designed for this type of problem? At least, the authors should consider this as an important future work.

Correctness: Yes. As far as I am concerned, I didn't spot any incorrectness in the method itself.

Clarity: Yes. The paper is well structured and easy to read. All the information needed seems to be in place.

Relation to Prior Work: Yes. The connections and differences between this work and previous works are well discussed.

Reproducibility: Yes

Additional Feedback:


Review 3

Summary and Contributions: The paper presents a supervised metric learning approach which leverages proxies. The method leverages proxies to better characterize the distribution of the entire dataset to address the sampling problem. The method formulates the proxies in a graph structure, along with a novel reverse label propagation algorithm. The authors conduct extensive experiments with their method including in the supplementary material and demonstrate superiority compared to numerous baselines.

Strengths: The authors explain their method in detail and provide numerous results demonstrating the benefit of using proxies in their approach. For example, they demonstrate that not only is retrieval / clustering performance improved, they show that their approach causes Faster convergence while obtaining higher-recall. The proposed method features some technical novelty. Specifically, leveraging a similarity graph over the N proxies in order to trade off between number of proxies (higher overhead / overfitting) vs too little information. Similarly, the authors use the similarity graph to construct sub-graphs to better capture local relationships around points. These sub-graphs then leverage a novel reverse label propagation technique. Most label propagation is used to predict missing labels from data from a learned semantic space, but in the authors case, they show how complete label information can be used to change the space (i.e. reverse propagation). While many metric learning approaches use labels to change the manifold, the use of the labels this way in sub-graphs to change the space seems to be of some novelty. Each component of the proposed method is explained in sufficient detail. Extensive experiments are provided for each parameter showing the effects of the parameter on recall. Similarly, ablation studies are present demonstrating the importance of the masks and masked softmax function, as well as the proxy regularization component. The method is compared with numerous other state-of-the-art metric learning approaches. Given the numbers are fairly close in many of the tables, highlighting based on statistical significance could have been helpful to determine whether results are significant for some methods (e.g. 86.3 vs 86.4 / 96.2 vs 96.0 / 90.6 vs 90.5). On CUB and Cars the method does seem to perform better however.

Weaknesses: I am a bit concerned about some of the claims in the paper being too strong. For example, the authors state at L67 that, "To our best knowledge, this is the first work that introduce graph classification into DML." However, I would be very careful making claims like this. There are many approaches in metric learning now that do similar things. For example: Li, Xiaocui, et al. "Semi-supervised clustering with deep metric learning and graph embedding." World Wide Web 23.2 (2020): 781-798. This approach is quite similar to yours actually. They use graphs as well as label propagation for deep metric learning, somewhat along the lines of what you do. It would have been nice to compare against this paper. However, at a minimum you could cite it and tone down the claims of being the first to use graph classification for DML. Writing quality needs improvements (see below). The method is supervised in that it requires class labels for data in order to learn the subgraph maps / proxies / etc. It would be nice to see how this method could be applied in "classless" settings, like vision+language settings where there is no concept of a class. This is more an idea for future work, however. Still, this limits the applicability of your method when class labels are not available. I would have liked to see how well this method works on larger-scale datasets. As the authors note, in the SOP dataset, there are sometimes around 5 instances per class. It would be nice to see how this could be expanded to much o I noticed that you use Inception pretrained on Imagenet. Do all the baselines also use that backbone? If not, it is possible some of your performance gain is coming from a more robust backbone architecture. It would be nice to indicate what the backbone is for each method, and if the backbone is different for those methods, to use your method with that backbone, especially for the most competitive methods. I noticed that the authors use the numbers from prior papers - while this is fine, if one is changing the backbone architecture, it is possible that the performance gains are not so much as claimed from your method, but rather from change of architecture. Have the authors considered imposing a diversity constraint among the proxies? For example, it is possible that the proxies collapse to a single point, or, that the proxies are not diverse across the class. Could a diversity constraint be imposed to enforce that the proxies are not too overlapping? Perhaps this was the purpose of the proxy regularization section, but if so, it is not clear to me why this occurs. I also recommend the authors to clarify that and make it explicit if that is the purpose. Despite being thorough, I am a bit concerned about the experimental evaluation. It would have been good to compare against recent approachies that are competitive that use proxies. For example: Aziere, Nicolas, and Sinisa Todorovic. "Ensemble deep manifold similarity learning using hard proxies." Proceedings of the IEEE Conference on Computer Vision and Pattern Recognition. 2019. also This was published in CVPR this year, so after the submission deadline, but you can still cite it: Kim, Sungyeon, et al. "Proxy Anchor Loss for Deep Metric Learning." Proceedings of the IEEE/CVF Conference on Computer Vision and Pattern Recognition. 2020. I would also note that this above method - Proxy Anchor loss is a drop in loss and has been available on Github for 4 months. https://github.com/tjddus9597/Proxy-Anchor-CVPR2020 It isn't required to cite it since wasn't published yet, but I recommend the authors add stronger proxy baselines. ProxyNCA (2017) is widely known as a weak method nowadays. Similarly, other baselines like clustering [13] (2017) and lifted structure loss [14] (2016) are dated and semihard [18] (2015) are dated. Your method leverages graph networks which may involve introducing additional parameters into the learning process. Thus, even if a prior baseline method uses the Inception baseline, your method has additional parameters - and is thus potentially more powerful. It would have been nice to tease this out. I have seen other papers control for parameter size, limit new parameters, or introduce additional layers into baseline methods to compensate. It is thus difficult to determine whether the gains are coming from your method or the increased parameters, or switch in backbone. The gain on Stanford Products is questionable, if any. Statistical tests are necessarily / useful to determine any increase in performance. ----------------------------------- POST REBUTTAL ----------------------------------- Please see my comments in feedback section - which address the weaknesses described above.

Correctness: The ablations / evaluations experiment with each component of the method and demonstrate improved performance. Each component of the method is demonstrated to result in increased performance. The experimental methodologies are reasonable and what one would expect. However, I am a bit concerned about a possible shift in baselines.

Clarity: There were also concerns about the writing quality throughout the paper. For example, even in the abstract, there are grammatical mistakes (e.g. "which uses fewer proxies while achieves better performance.") I would recommend the authors do a thorough read-through this paper to correct such writing errors. I was also lost at times during the explanation of the method. It was not immediately clear to me what the proxy regularization loss was doing. It seems to just be a classificaiton loss? This should be clarified in the text.

Relation to Prior Work: The authors include related work and some discussion of the differences of their work with other proxy methods. However, given the overly strong claims made, I believe the authors may have not been aware of other graph methods for metric learning. I recommend the authors clearly differentiate their work from: Li, Xiaocui, et al. "Semi-supervised clustering with deep metric learning and graph embedding." World Wide Web 23.2 (2020): 781-798. I also recommend they include a discussion with some of the papers I cited above.

Reproducibility: Yes

Additional Feedback: Overall, I think this is a good submission - but there are significant concerns about experimental evaluation. Most significantly, I am concerned about the lack of comparison to recent proxy based methods. The significance of the method on SOP dataset is questionable / if any. Similarly, it is difficult to assess the contribution of the method due to change in backbone / extra parameters / etc. If authors can verify baselines for SoftTriple and MS are the same, then maybe add the CVPR 2020 or the CVPR2019 paper cited above as an additional proxy-based baseline, I feel the paper would be strong enough for acceptance. Even though the CVPR2020 one wasn't published at the time, the CVPR2019 one was. As it stands, there are many proxy-based deep metric learning approaches which the authors have failed to compare to. ----------------------------------- POST REBUTTAL ----------------------------------- I was happy to see that the authors have taken into account the feedback I suggested above. In particular, they added a comparison to a CVPR2020 paper that also used proxies as I suggested. Their method significantly outperforms this method as shown in their rebuttal. They also discussed that the backbones of the baselines they compared against were the same. They state they will include this in their final submission, which I strongly encourage they do (and is somewhat typical nowadays). This ensures a fair comparison is made. They also discussed their performance on SOP to my satisfaction. At this time, my concerns are satisfied. This paper presents a solid contribution as a NeurIPS paper and I believe should be accepted.

[Author Response · NeurIPS 2020]

We thank the reviewers for their valuable comments and recognition of the novelty and results of our method, *e.g.*, this
work is well structured and motivated [R4] / explained in sufficient detail [R5]; the results show improvements [R2 R5] /
effectiveness [R4]; experiments and ablation studies are sound [R2] / comprehensive [R4] / reasonable [R5]; "I'm
positive to see such a novel idea" [R4]; "the proposed method features some technical novelty" [R5]; "I think this is a
good submission" [R5]. We respond to the major comments below but will address all feedback in our revised version.

[R2] **Roles of $S^{pos}$, graphs and proxies.** $S^{pos}$, together with proxies, graphs, and all other modules, are designed to
learn a better embedding network (in our case, the Inception backbone), and will be totally removed during testing, like
ProxyNCA [12]. During training, the global graph $S$ is used to construct $k$-NN sub-graphs $W$, and immediately $W$
serves for the proposed *reverse label propagation* in Eq. (6), during which the label message of proxy nodes will be
passed to the sample nodes with a certain probability score according to $W$. Thus, graphs are indispensable in our
method, and constructing $k$-NN sub-graphs is conducive to learning discriminative proxies by focusing more on the
local manifold of each sample. Proxies are globally learnable "cluster centers" while Clustering [13] directly regards
input samples as cluster centers, and we will further clarify the differences between our method and other approaches.

[R2 R5] **Constraints among proxies.** There are actually two types of constraints among proxies in our method, *i.e.*, a
"soft" constraint, by encouraging proxies to be close to their anchor samples (**L135–143**), and a "hard" one, by forcing
similar proxies to be close to each other while dissimilar ones far away from each other (**L192–195**). The soft constraint
allows capturing diverse intra-class differences while the hard one concentrates on the discrimination of the embedding
space. In practice, similar proxies tend to be sufficiently close to each other in the later training stage. We argue that,
during training, our model dynamically trades off between diversity and discrimination to achieve better performance.

[R4] **Selecting $N$ and $r$.** Empirically, a large $N$ is suitable for datasets with large intra-class variance while a small
sub-graph (with $r = 0.05$) is optimal in most cases. Accordingly, the effects of $N$ and $r$ follow similar patterns on
CUB as on Cars196 (both with $N = 12$); for SOP dataset the optimal case is $N = 1$ due to its low intra-class variance.

[R4 R5] **Performance on SOP.** Though our method does not consistently outperform the most competitive baselines on
SOP (11318 classes) under all metrics, it achieves comparable results with a much less computational cost. Specifically,
MS [24] adopts a very large batch size 1000 (see its appendix) to achieve its best performance, which is difficult for us
to reproduce even using four GPUs, each with 12 GB memory. Proxy-Anchor (CVPR 2020) also requires a large batch
size 180 and calculates each sample with all 11318 proxies; SoftTriple [15] employs two parallel FC layers to classify
11318 classes. In contrast, our method only needs to calculate and update the gradients of $\lceil 0.05 \times 11318 \times 1 \rceil$ (see
Eq. (5)) proxies for each sample during back-propagation, and we use a small batch size 32 for *inheriting the advantage*
of original ProxyNCA. As future work, we will focus more on addressing such datasets with huge inter-class variance.

[R5] **Backbone and parameters.** We use the same backbone as in MS [24] and SoftTriple [15] — Inception pretrained
on ImageNet; actually we strictly follow the setting of SoftTriple, *i.e.*, data pre-processing, backbone (followed by
an average pooling layer), optimizer, *etc*. We will follow the suggestion to indicate the backbones used by other
methods. Regarding the concern about whether the performance improvement is brought by additional parameters in
the graphs, actually only proxies are trainable parameters — we do not involve any extra conv, FC layers or other forms
of parameters. In fact, our method contains a similar number of parameters to SoftTriple while performing better in
most cases; besides, only $\frac{1}{20}$ of these parameters are adaptively calculated and updated during each back-propagation.
Therefore, we confirm that our performance improvement is not caused by switch in backbone or increased parameters.

[R5] **More comparisons.** Following the suggestion, we compare our ProxyGML
with the mentioned Proxy-Anchor (CVPR 2020) using its official code, and the
Recall@1 results are shown in the table. In particular, we have found that Proxy-
Anchor relies on a large batch size, and is implemented with three additional
engineering skills, *i.e.*, 1) a combination of an average- and a max- pooling layer
following the Inception backbone, 2) a warm-up strategy for stabilizing proxy
learning, and 3) an AdamW optimizer instead of the original Adam. For fair
comparison, we evaluate Proxy-Anchor under our setting — with batch size 32

| Method | CUB | Cars196 | SOP |
|---|---|---|---|
| ProxyGML$_{32}$ | **66.6** | **85.5** | **78.0** |
| Proxy-Anchor$_{32}$ | 35.8 | 20.3 | 41.4 |
| Proxy-Anchor$^*_{32}$ | 65.4 | 83.1 | 75.7 |
| Proxy-Anchor$_{180}$ | 66.1 | 84.2 | 54.5 |
| Proxy-Anchor$^*_{30}$ | 65.9 | 84.6 | 76.0 |

and the three engineering skills removed; it is also evaluated with the three skills enabled (indicated by "*"), and with
its optimal batch size 180. Since the time does not allow any further tuning for Proxy-Anchor, we report here the result
with batch size 30 (also the skills are used) provided in its paper for reference. We note that this is only a preliminary
experiment, and in the revised version we will further involve more experiments of Proxy-Anchor with careful tuning,
and ProxyGML with large batch size and the three skills added. Still, we can infer from the table the advantage of our
ProxyGML against Proxy-Anchor. Considering that the mentioned EDMS (CVPR 2019) is practically an extension of
N-pair loss with heavy backbone ensembles and has no code released, we do not involve it for comparison (same as
Proxy-Anchor), but will add it for discussion. Also, we will tone down the claims of being the first to use graph
classification for DML and clarify the difference between the mentioned SCDMLGE (WWW 2020) and our ProxyGML.

[Meta-Review · NeurIPS 2020]

The paper proposes a relevant and novel idea and the experiments are comprehensive. The technical contribution is strong. The paper is well written and clear. After the author response, where an important missing comparison to the state of the art was provided, all reviewers agree on accept.